# Generating Post-Acetazolamide Cerebral Blood Flow MRI for High-Risk Stroke Patients

**Rydham Goyal**[*1]                                                    RYDHAM@STANFORD.EDU

**Camila Gonzalez**[2]                                                CAMGONZA@STANFORD.EDU

**Sasha Alexander**[3]                                                SASHALEX@STANFORD.EDU

**Aja Zou**[3]                                                              AJAZOU@STANFORD.EDU

**Michael E Moseley**[2]                                            MOSELEY@STANFORD.EDU

**Moss Y Zhao**[†3]                                                    MOSSZHAO@STANFORD.EDU

**Gary K Steinberg**[‡3]                                            CEREBRAL@STANFORD.EDU

[1] *Department of Computer Science, School of Engineering, Stanford University, 450 Jane Stanford Way, Stanford, CA 94305*

[2] *Department of Radiology, Stanford University School of Medicine, 300 Pasteur Drive, Stanford, CA 94305*

[3] *Department of Neurosurgery, Stanford University School of Medicine, 300 Pasteur Drive, Stanford, CA 94305*

**Editors:** Accepted for publication at MIDL 2026

## Abstract

Cerebrovascular reserve (CVR) quantifies the brain's ability to augment cerebral blood flow in response to a vasodilatory stimulus. It is a key biomarker in Moyamoya disease and other steno-occlusive cerebrovascular disorders. Clinically, CVR is typically assessed by administering acetazolamide (ACZ) and acquiring post-ACZ perfusion maps, but this workflow is time-consuming, costly, and contraindicated in a subset of patients. In this work, we investigate whether deep learning can predict post-ACZ perfusion directly from baseline arterial spin labeling (ASL) MRI, enabling pharmacologic-free CVR estimation. We curate a single-center dataset of Moyamoya ASL perfusion imaging, comprising pre/post-ACZ scan pairs from 194 patients. We design a post-ACZ conditional Autoencoder (cAE) network to regress the middle axial post-ACZ slice from the corresponding pre-ACZ slice using a combined L1 and SSIM loss. We evaluate our method against three diffusion-based formulations (conditional DDPM, Cold Diffusion, and Residual Diffusion). On a holdout test set of 49 patients, the proposed post-ACZ cAE achieves the highest reconstruction fidelity (SSIM $\approx 0.79$), outperforming diffusion-based baselines in MAE, SSIM, and PSNR. Region-wise analysis of CBF percentage change in affected versus healthy MCA territories showed that the generated post-ACZ model outputs followed ground truth patterns of cerebrovascular reserve. Our findings demonstrate the feasibility of non-invasive CVR assessment using MRI for high-risk patients. Our data-driven approach could reduce reliance on ACZ challenges in routine clinical workflow and expand access to CVR testing to evaluate brain health.

**Keywords:** Generative models, Conditional Autoencoder, Pixel-space diffusion, Image-to-image translation, Arterial Spin Labeling, MRI, Cerebrovascular reserve

---

[*] First author

[†] Co-senior author

[‡] Senior author

## 1. Introduction

Cerebrovascular Reserve (CVR) reflects the brain's ability to increase cerebral blood flow (CBF) in response to a vasodilatory stimulus and serves as a critical biomarker in evaluating brain health and the risk for cerebrovascular diseases (Kancheva et al., 2025). Clinically, CVR is typically assessed using a vasoactive challenge, such as the administration of acetazolamide (ACZ, Diamox), during medical imaging exams (Zhao et al., 2022b). In standard practice, repeated CBF imaging techniques, such as arterial spin labeling (ASL) MRI, are performed before and after intravenous ACZ administration (Fahlström et al., 2021). However, these procedures have significant limitations: they are time-intensive, costly, and occasionally contraindicated due to ACZ side-effects (Schmickl et al., 2020). As a result, there is strong clinical motivation to develop non-invasive, computational methods that can predict post-ACZ CBF responses directly from baseline (pre-ACZ) imaging, thereby reducing the need for pharmacological challenges.

Recent advances in medical image-to-image translation have enabled data-driven approaches to synthesize new images conditioned on existing data (Bi, 2023). For example, deep learning models, such as encoder–decoder architectures, have demonstrated promise in modeling nonlinear mappings between imaging data acquired under different conditions, transforming pre-ACZ to post-ACZ hemodynamic states (Chen et al., 2020). Yet these models remain limited by their deterministic nature: given a single input, standard encoder-decoder models always produce a single fixed output. This setting does not fully represent clinical workflow, as repeated acquisitions of the same patient ostensibly under identical physiological conditions naturally exhibit stochastic variability due to noise, patient movement, and physiological fluctuations. A purely deterministic mapping, therefore, underrepresents the inherent uncertainty in both biological and measurement processes. In contrast, diffusion-based generative models sample from a learned distribution rather than producing a single deterministic output (Kazerouni et al., 2023). Unlike deterministic models, diffusion models can generate multiple plausible post-ACZ realizations from the same pre-ACZ imaging, better capturing natural physiological variability.

Predicting post-ACZ imaging using generative methods could reduce reliance on ACZ injection while providing uncertainty-aware post-ACZ realizations. In this work, we propose a post-ACZ conditional autoencoder (cAE) as our primary model for generating synthetic post-ACZ CBF maps directly from baseline ASL in Moyamoya disease. While we also implement several diffusion-based generative models (a conditional DDPM, Cold Diffusion, and a residual diffusion variant) to explore the potential benefits of more complex probabilistic mappings, our emphasis is on demonstrating that a simpler, data-efficient encoder–decoder architecture can achieve superior performance and is easier to train and deploy on limited clinical datasets. We evaluate all approaches on paired pre/post-ACZ scans using image-level reconstruction metrics—Structural Similarity Index (SSIM), Peak Signal-to-Noise Ratio (PSNR), and Mean Absolute Error (MAE)—together with MCA territory ROI analyses of percentage CBF change, which together show that the proposed cAE not only attains the best quantitative reconstruction scores but also most faithfully preserves clinically meaningful cerebrovascular reserve patterns.

## 2. Related Work

Recent research on non-invasive CVR assessment has highlighted the strengths of using deep learning in predicting vasodilatory responses directly from imaging data collected before vasodilation, reducing the need for acetazolamide (ACZ) challenges or gold-standard but logistically demanding PET imaging (Puig et al., 2020). Encoder–decoder models have shown that voxel-wise CVR can be estimated from multimodal MRI inputs, with studies demonstrating substantially improved agreement with PET-derived CVR compared with conventional ASL-based calculations (Chen et al., 2020). Similar work in multimodal synthesis has generated PET CBF maps from structural and perfusion MRI, including recent 3D architectures with attention mechanisms that achieve high fidelity and strong diagnostic performance (Guo et al., 2020) (Hussein et al., 2024). While these approaches highlight the potential of learning nonlinear mappings between baseline CBF and physiological hemodynamic states, they are deterministic, thus generating only a single predicted outcome for each input; they are unable to capture the intrinsic biological and measurement variability observed across repeated perfusion acquisitions (Teo et al., 2022).

Similarly, advances in generative modeling, particularly diffusion models, have enabled high-resolution, anatomically consistent synthesis of 3D medical images and offer a principled framework for sampling from full conditional distributions rather than fixed point estimates (Kazerouni et al., 2023). Complementary research comparing single- versus multi-PLD ASL, and studies documenting ATT-related biases during ACZ-induced vasodilation, further underscore the complexity of CVR and the need for models that respect regional physiologic constraints (Zhao et al., 2022a). These developments motivate our probabilistic diffusion-based framework for synthesizing post-ACZ CBF, extending deterministic prediction toward a more realistic generative representation of cerebrovascular physiology.

From a modeling perspective, our approach builds on three main streams of prior research. First, deep learning has already been used to cast CVR mapping itself as an image-to-image prediction problem: Chen et al. showed that convolutional networks can recover voxel-wise CVR maps from baseline or resting-state MRI without explicit hypercapnic challenge, providing a direct precedent for estimating vasodilatory responses from pre-challenge imaging alone (Chen et al., 2020). Second, our deterministic baseline follows the U-Net encoder–decoder architecture, whose multi-scale feature hierarchy and skip connections have become the standard backbone for biomedical image-to-image tasks and underlies many recent CVR and perfusion prediction models (Ronneberger et al., 2015). Third, we draw on the emerging literature on diffusion-based generative modeling in medical imaging: Pinaya et al. demonstrated that denoising diffusion probabilistic models can generate high-resolution 3D brain images with realistic anatomical variability (Pinaya et al., 2022), while Özbey et al. extended diffusion to unsupervised medical image translation, using a source modality to guide the reverse diffusion process for cross-contrast synthesis (Özbey et al., 2023). Complementary to these noisy formulations, Bansal et al. introduced Cold Diffusion, replacing Gaussian noise with deterministic degradations and learning to invert them via iterative refinement, thereby providing a noise-free diffusion framework well suited to structured inverse problems (Bansal et al., 2023). Our methodology adapts these ideas to the clinically motivated setting of the acetazolamide challenge by combining both a MONAI-based convolutional encoder with conditional DDPM and Cold Diffusion

formulations tailored to the pre- to post-ACZ mapping, thereby shifting the problem from deterministic regression to explicitly modeling the conditional distribution of post-ACZ perfusion given baseline ASL.

## 3. Materials and Methods

### 3.1. Data Preparation

Our data comprises paired baseline (pre-acetazolamide) and post-acetazolamide (post-ACZ) cerebral blood flow (CBF) ASL MRI scans acquired from patients with cerebrovascular disease over several decades at Stanford University. Detailed acquisition and reconstruction information has been reported previously (Zhao et al., 2023). Each pair consists of co-registered ASL-based CBF images, standardized to a uniform voxel size and slice orientation. A total of 194 paired scan volumes were available for this study. The dataset was split into training (63%), validation (12%), and test (25%) sets, resulting in 127 training, 18 validation, and 49 test samples. The random split was performed with a fixed seed to ensure reproducibility across all experiments.

We generate a single mid-axial slice because, in Moyamoya disease, cerebrovascular pathology predominantly involves the anterior circulation, particularly the bilateral MCA territories, which are consistently intersected by the mid-axial plane. In practice, CVR interpretation is typically performed at the vascular-territory level rather than requiring voxel-complete 3D coverage. Methodologically, restricting to a 2D slice improves data efficiency and model stability in a limited paired-data regime (194 paired scans), enabling a rigorous feasibility test of whether post-ACZ hemodynamic changes are predictable from baseline ASL. The slice is resized to $128 \times 128$ and min–max normalized to $[0, 1]$.

A vascular territory atlas with $2\,\text{mm}$ resolution was incorporated to define regions of interest (ROIs) corresponding to the left and right Middle Cerebral Artery (MCA) territories. For each patient, the disease side (left, right, or bilateral) was annotated, enabling region-wise evaluation of model predictions against ground truth post-ACZ scans.

### 3.2. Model Architectures

We evaluated four distinct approaches for predicting post-ACZ CBF maps from pre-ACZ inputs.

### 3.2.1. POST-ACZ CONDITIONAL AUTOENCODER (CAE)

The post-ACZ conditional Autoencoder (cAE) serves as our primary supervised baseline for predicting post-ACZ CBF from baseline ASL. We implement the cAE as a fully convolutional 2D U-Net using the MONAI library, with a single-channel pre-ACZ slice as input and a single-channel post-ACZ slice as output. The network follows a four-level encoder–decoder structure: the encoder progressively downsamples the input via strided convolutions, increasing the number of feature channels at each resolution level, while the decoder symmetrically upsamples back to the original spatial resolution. Skip connections between encoder and decoder layers propagate fine-grained spatial information forward, enabling the model to preserve detailed anatomy while learning nonlinear transformations associated with the vasodilatory response. Each block comprises residual units with LeakyReLU activations and

instance normalization, which empirically stabilized optimization and improved convergence in our setting.

Conceptually, we interpret this architecture as a conditional autoencoder in which the latent representation encodes a compact description of the baseline perfusion pattern together with the information needed to synthesize the corresponding post-ACZ state. Unlike an unconditional autoencoder that reconstructs its own input, the cAE is trained to map from pre-ACZ to post-ACZ images, thereby explicitly learning the treatment-induced change while still leveraging the strong inductive biases of U-Net–style multi-scale feature hierarchies. We selected this design because U-Net variants are the de facto standard for medical image-to-image translation tasks and have previously been shown to perform well for CVR and perfusion prediction from multimodal MRI.

Training is performed end-to-end using a composite objective that balances voxelwise fidelity and perceptual structural similarity. Given a pre-ACZ slice $x_{\mathrm{pre}}$ and its corresponding post-ACZ slice $x_{\mathrm{post}}$, the cAE predicts $\hat{x}_{\mathrm{post}} = f_\theta(x_{\mathrm{pre}})$ and the loss is defined as

$$\mathcal{L}(\theta) = \mathcal{L}_{L1}\big(\hat{x}_{\mathrm{post}}, x_{\mathrm{post}}\big) + \big(1 - \mathrm{SSIM}(\hat{x}_{\mathrm{post}}, x_{\mathrm{post}})\big),$$

where $\mathcal{L}_{L1}$ denotes the mean absolute error and SSIM is the structural similarity index computed over the 2D slice. The $L1$ term encourages accurate intensity regression of CBF values, while the SSIM term promotes preservation of local contrast and anatomical structure. We optimize this objective with the Adam optimizer, using a mini-batch size of 8 and an initial learning rate of $10^{-3}$. For hyperparameter selection, we held out 25% of subjects as a fixed test set (seed 1337). On the remaining data, we selected cAE hyperparameters using 3-fold cross-validation (shuffled; seed 1337), scoring each fold by the best validation SSIM achieved during training. We evaluated a small grid over learning rate and U-Net channel width (3 total configurations): learning rate $\in \{10^{-3}, 5 \times 10^{-4}\}$ and channel tuples $\in \{(16, 32, 64, 128), (32, 64, 128, 256)\}$. For each fold/configuration, we trained for 10 epochs (batch size 8; Adam) using the composite objective $\mathcal{L} = \mathcal{L}_1 + (1 - \mathrm{SSIM})$, and selected the configuration with the highest mean validation SSIM across folds.

### 3.2.2. Diffusion-based alternative formulations

To contextualize the performance of the proposed post-ACZ cAE, we also investigated three diffusion-based alternative formulations of the pre-to-post mapping. All three operate on the same pre-/post-ACZ pairs and middle-slice preprocessing as the cAE, but differ in how they formulate the generative task and incorporate stochasticity.

First, we implemented a deterministic Cold Diffusion model in which corruption is defined by linear interpolation between the target post-ACZ slice and the observed pre-ACZ slice. For each pair $(x_{\mathrm{pre}}, x_{\mathrm{post}})$, we generate intermediate states $x_t = (1 - \alpha_t)x_{\mathrm{post}} + \alpha_t x_{\mathrm{pre}}$ over $T = 10$ steps, with $\alpha_t$ increasing from 0 (clean post-ACZ) to 1 (approximately pre-ACZ) using an interpolation schedule $\alpha_t = t/(T - 1)$. A U-Net restoration network (MONAI 2D `UNet` with in_channels= 2, out_channels= 1, channels= $(64, 128, 256, 512)$, strides= $(2, 2, 2)$, and num_res_units= 2, using instance normalization and LeakyReLU activations) receives the degraded slice $x_t$ concatenated with $x_{\mathrm{pre}}$. It is trained to predict $x_{\mathrm{post}}$ for 50 epochs (batch size 8) with Adam (lr $10^{-3}$) using the same composite $L1 + (1 - \mathrm{SSIM})$ loss as the cAE. At inference, we run this process in reverse, starting from $x_{T-1} \approx x_{\mathrm{pre}}$ and iteratively

refining toward a synthetic post-ACZ image, with a simple foreground mask derived from $x_{\mathrm{pre}}$ (threshold 0.05) used to preserve background intensities.

Second, we considered a fully stochastic conditional DDPM that learns a distribution over post-ACZ slices given the pre-ACZ slice. Here, Gaussian noise is added to $x_{\mathrm{post}}$ over $T = 200$ diffusion steps with a linear variance schedule $\beta_t \in [10^{-4}, 2 \times 10^{-2}]$, and a MONAI `DiffusionModelUNet` (2D; in_channels= 2, out_channels= 1; num_channels= $(64, 128, 256)$; attention_levels= (`False`, `True`, `True`); num_res_blocks= 2; num_head_channels= $(0, 32, 32)$; norm_num_groups= 8) is trained to predict the injected noise from the noisy image and $x_{\mathrm{pre}}$ using an MSE objective with Adam (lr $2 \times 10^{-4}$) for 50 epochs (batch size 8). This noise-prediction objective yields a time-indexed family of denoising operators conditioned on baseline perfusion, from which we generate a post-ACZ sample by running the reverse process from pure noise.

Finally, we explored a residual diffusion formulation that focuses explicitly on the ACZ-induced change $r_0 = x_{\mathrm{post}} - x_{\mathrm{pre}}$ rather than the full post-ACZ anatomy. The residual is scaled to a roughly symmetric range via $r_0 \times 2.0$, diffused forward with a DDPM-style schedule over $T = 100$ steps (linear $\beta_t \in [10^{-4}, 2 \times 10^{-2}]$), and a UNet-based network (the same MONAI 2D `UNet` architecture as the Cold Diffusion baseline) is trained to denoise the residual while conditioned on $x_{\mathrm{pre}}$ using an MSE objective with Adam (lr $10^{-3}$) for 50 epochs (batch size 8). At test time, we sample a denoised residual and add it back to the baseline image to obtain a post-ACZ prediction. This residual view is conceptually appealing as it isolates the treatment effect; its empirical performance relative to the cAE and other diffusion-based formulations is reported in Section 4.

### 3.3. Evaluation Metrics

To assess how well each model recovered post-ACZ perfusion, we used three complementary image-level metrics: the Structural Similarity Index (SSIM), Peak Signal-to-Noise Ratio (PSNR), and Mean Absolute Error (MAE). MAE provides a direct measure of voxel-wise intensity disagreement between the predicted and ground-truth post-ACZ slices by averaging the absolute difference across all pixels. This error measure is easy to interpret in the native CBF intensity scale and is sensitive to global under- or overestimation of perfusion. SSIM, in contrast, is designed to capture perceptual and structural fidelity rather than purely point-wise differences. It compares local patterns of luminance, contrast, and structure between the two images, yielding a value in $[0, 1]$ where higher scores indicate better preservation of anatomical detail and regional perfusion patterns. PSNR complements these metrics by expressing the ratio of signal power to reconstruction error on a logarithmic decibel scale, derived from the mean squared error. Higher PSNR values correspond to lower noise-like deviations and better overall image quality. Therefore, these three metrics allow us to simultaneously quantify absolute intensity accuracy, structural similarity, and noise behavior in the synthesized post-ACZ maps.

In addition to global image-level scores, we performed a region-wise analysis tailored to cerebrovascular reserve. For each patient, we computed the percentage change in CBF within the left and right MCA territories defined by the vascular atlas. At the voxel level,

| Model | MAE $\downarrow$ | SSIM $\uparrow$ | PSNR (dB) $\uparrow$ |
|---|---|---|---|
| Cold Diffusion | $0.0660 \pm 0.0260$ | $0.7195 \pm 0.0920$ | $18.66 \pm 2.39$ |
| Diffusion (DDPM) | $0.0841 \pm 0.0243$ | $0.4486 \pm 0.0800$ | $17.33 \pm 2.04$ |
| Residual Diffusion | $0.1976 \pm 0.0094$ | $0.0863 \pm 0.0215$ | $11.25 \pm 0.38$ |
| post-ACZ cAE | $\mathbf{0.0497 \pm 0.0176}$ | $\mathbf{0.7886 \pm 0.1135}$ | $\mathbf{21.49 \pm 2.70}$ |

Table 1: Quantitative performance of all models on the held-out test set (49 patients). Values reported as mean $\pm$ standard deviation.

the percentage change was defined as

$$\Delta\mathrm{CBF}_i = \frac{I_{\mathrm{post},i} - I_{\mathrm{pre},i}}{I_{\mathrm{pre},i}} \times 100\%,$$

where $I_{\mathrm{pre},i}$ and $I_{\mathrm{post},i}$ denote the baseline and post-ACZ intensities at voxel $i$, respectively. We then summarized the territorial response by averaging over all valid voxels with nonzero baseline perfusion ($I_{\mathrm{pre},i} > 0$),

$$\overline{\Delta\mathrm{CBF}} = \frac{1}{N} \sum_{i=1}^{N} \Delta\mathrm{CBF}_i,$$

yielding a single percentage change value per territory and per subject. We clip voxelwise $\Delta\mathrm{CBF}_i$ to the range $[-200\%, 200\%]$ before computing territorial averages. We do not threshold low baseline voxels beyond requiring $I_{\mathrm{pre},i} > 0$, since very low baseline perfusion can reflect severe hypoperfusion and excluding these voxels would remove clinically relevant regions. This analysis probes whether the models reproduce clinically meaningful patterns of vasodilatory response, such as attenuated augmentation in diseased MCA territories compared to contralateral or healthy regions. We note that this voxelwise normalization can be sensitive to very low baseline intensities, but it provides a physiologically interpretable measure of relative CBF augmentation. Finally, all regional summaries were stratified by disease laterality (left, right, bilateral) to evaluate how model performance varied across different patterns of cerebrovascular involvement.

## 4. Results and Discussion

Quantitative reconstruction performance on the held-out test cohort is summarized in Table 1. The proposed post-ACZ conditional autoencoder (cAE) achieves the best overall fidelity, with the lowest mean absolute error (MAE; $0.0497 \pm 0.0176$), the highest structural similarity index (SSIM; $0.7886 \pm 0.1135$), and the highest peak signal-to-noise ratio (PSNR; $21.49 \pm 2.70$ dB). Together, these metrics indicate that the cAE most accurately preserves both voxelwise intensities and fine-grained anatomical structure in the synthesized post-ACZ CBF maps.

Among the diffusion-based models, the Cold Diffusion model performs second best, with MAE $0.0660 \pm 0.0260$, SSIM $0.7195 \pm 0.0920$, and PSNR $18.66 \pm 2.39$ dB. Although competitive, it consistently underperforms the cAE across all three metrics, suggesting

that inverting a deterministic interpolation process is less effective than direct conditional regression for this dataset. The DDPM-based diffusion model exhibits higher reconstruction error (MAE $0.0841 \pm 0.0243$) and lower SSIM ($0.4486 \pm 0.0800$) and PSNR ($17.33 \pm 2.04$ dB), reflecting blurrier and noisier reconstructions when sampling from the learned conditional distribution. The residual diffusion formulation performs markedly worse than all other approaches, with MAE close to 0.20, SSIM near zero, and PSNR around 11 dB, indicating that modeling only the residual perfusion change with a stochastic diffusion process fails to recover anatomically plausible post-ACZ images in this setting.

The superior performance of the deterministic cAE over the stochastic diffusion formulations offers an important insight into the nature of the hemodynamic response in ASL. While we hypothesized that modeling the stochastic variability of perfusion would improve realism, our results suggest that the mapping from pre- to post-ACZ states in Moyamoya disease is sufficiently constrained to be modeled effectively by direct regression. The diffusion models, while theoretically capable of capturing multi-modal distributions, instead appeared to introduce high-frequency variance that degraded voxel-wise fidelity (PSNR/SSIM) without yielding a perceptible benefit in anatomical plausibility. This indicates that for ASL-based CVR assessment, where the primary clinical signal is a global or regional shift in mean intensity rather than high-frequency texture generation, the inductive bias of a U-Net trained with an L1/SSIM objective is better aligned with the task than the generative diversity of diffusion.

To assess whether these reconstruction differences translate into physiologically meaningful cerebrovascular reactivity (CVR) patterns, we also conducted a mask-based analysis of percentage CBF change. For each patient, we computed voxelwise $\Delta$CBF between pre- and post-ACZ images within atlas-defined middle cerebral artery (MCA) territories, and summarized mean percentage change separately for diseased and contralateral (unaffected) regions, stratified by disease laterality. Ground-truth measurements showed larger CBF increases in healthy territories than in diseased MCA regions, consistent with impaired vasoreactivity in steno-occlusive disease, but the distributions exhibited substantial variance due to pixel-level outliers in the percentage-change calculation. Figure 1 reports the cohort mean percentage change for diseased vs. contralateral MCA territories, stratified by disease laterality.

Within this CVR-focused evaluation, the post-ACZ cAE learned the clinically expected ordering of vascular responses across all three laterality groups (Figure 1). For left-sided disease ($n = 11$), ground truth showed larger CBF increases in healthy versus diseased MCA territories (67.41% unhealthy vs. 223.29% healthy), and the cAE preserved this pattern with a compressed dynamic range (30.60% vs. 69.83%). The same ordering was observed for right-sided disease (47.03% vs. 131.22% in ground truth, 7.10% vs. 97.19% for the cAE) and bilateral disease (136.38% vs. 1308.48% in ground truth, 42.78% vs. 206.72% for the cAE), indicating that the model consistently learned that healthy MCA territories should exhibit larger perfusion augmentation than their diseased counterparts.

In a subset of subjects, diseased territories showed negative mean $\Delta$CBF in both ground-truth and cAE-predicted maps, reflecting paradoxical decreases in CBF after acetazolamide. This "steal"-like behavior (paradoxical CBF reduction after vasodilatory challenge) is clinically meaningful, as it may help identify patients at particularly high risk of ischemia under vasodilatory stress. Nonetheless, the precision of these percentage changes is limited by

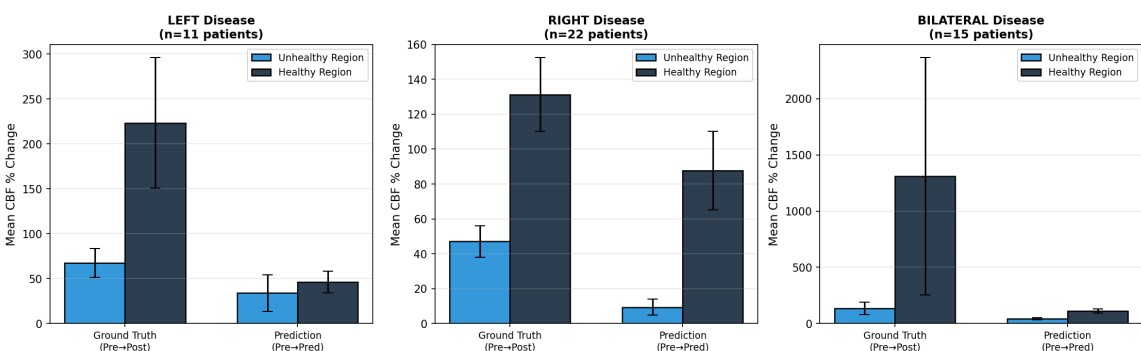

Figure 1: Mean percentage CBF change within atlas-defined MCA territory masks for ground truth (pre→post) and cAE prediction (pre→pred), stratified by disease laterality (left n=11, right n=22, bilateral n=15; total n=49). Error bars denote the standard deviation across subjects.

the modest dataset size and the inherently non-deterministic physiologic response to ACZ, and sensitivity to voxels with very low baseline CBF. Division by near-zero pre-ACZ intensities can generate extreme $\Delta$CBF values; although we applied a $\pm 200\%$ cap to mitigate the most severe outliers, large variability remains, especially in healthy bilateral territories. Future work should adopt more robust regional statistics (e.g., winsorization, IQR-based outlier handling, or median-based summaries) to better characterize territorial CVR while preserving genuine biological variability.

Finally, we highlight the single best-performing cAE prediction on the test cohort (Figure 2). This slice achieved the highest PSNR among all evaluated examples (25.91 dB), the lowest prediction error (MAE = 0.0289) within the top performers, and a SSIM of 0.8765, only slightly below the best SSIM case (0.8817). Notably, the ground-truth CBF change for this subject was relatively small (0.0232), and the cAE nevertheless captured this subtle augmentation accurately, preserving territorial structure while closely matching the absolute magnitude of the post-ACZ response.

## 5. Conclusion

In this work, we presented a proof-of-concept framework for synthesizing post-acetazolamide (post-ACZ) CBF maps directly from baseline ASL MRI in patients with Moyamoya and related steno-occlusive disease. Across a cohort of 194 paired pre/post-ACZ scans, our post-ACZ conditional Autoencoder (cAE) consistently achieved the best image-level reconstruction performance on a held-out test set, outperforming three diffusion-based models in MAE, SSIM, and PSNR. Beyond global fidelity, MCA territory analyses of percentage CBF change demonstrated that the cAE recovered the clinically expected ordering of responses—larger augmentation in healthy than diseased territories—and reproduced "steal"-like decreases in perfusion in high-risk regions, indicating that the model captures

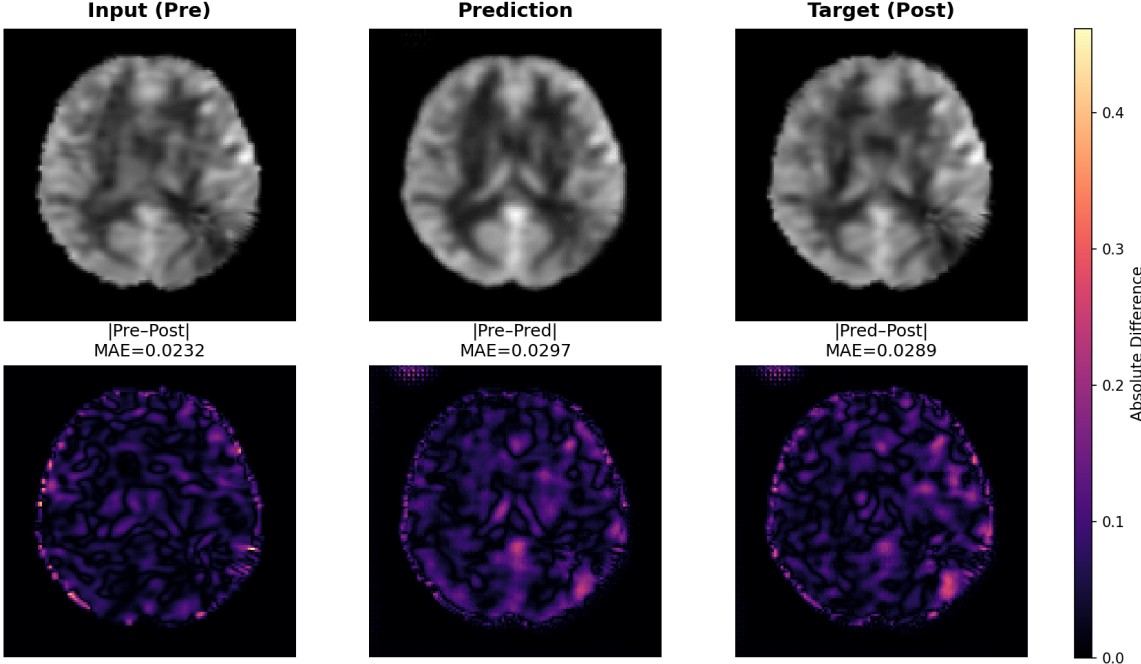

Figure 2: Best-performing post-ACZ prediction from the cAE on a held-out test slice. From left to right: pre-ACZ input, cAE-generated post-ACZ slice, and ground-truth post-ACZ perfusion.

physiologically meaningful cerebrovascular reserve patterns rather than merely minimizing pixelwise error.

These results suggest that a relatively simple, data-efficient encoder–decoder architecture can serve as a practical tool for non-invasive CVR estimation when only pre-ACZ ASL imaging is available, potentially reducing reliance on pharmacologic ACZ challenges in patients for whom they are risky, poorly tolerated, or operationally infeasible. At the same time, our findings highlight several important limitations. The current study is restricted to a single-center dataset and a single 2D middle axial slice per subject, and the voxelwise percentage-change analysis remains sensitive to low baseline CBF and outliers, even with capping. Future work should therefore extend this framework to full 3D volumetric modeling of the entire perfusion volume, incorporate more robust territorial summary statistics, and evaluate generalizability on larger, multi-center cohorts. In parallel, uncertainty-aware generative extensions, such as better-calibrated diffusion models tailored to this data regime, may provide complementary information about confidence in predicted CVR patterns. Collectively, these approaches advance the development of clinically deployable, pharmacologically-free CVR assessment pipelines that integrate seamlessly with routine MRI workflows for high-risk stroke patients.

## Acknowledgments

This work is supported in part by the American Heart Association Career Development Award 24CDA1266771 and the Stanford Maternal and Child Health Research Institute Pilot Grant. The authors thank Christine Plant and Jeanne Gu for manuscript editing and database management.

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
