# OpenReview forum: "Generating Post-Acetazolamide Cerebral Blood Flow MRI for High-risk Stroke Patients"
_MIDL.io/2026/Conference — MIDL 2026 Poster_

### Official Review · Reviewer_s8hq · 2026-01-09

**Confidence:** 3
**Preliminary Rating:** 3

**Summary:**

This study uses deep learning to estimate cerebrovascular reserve non-invasively by generating synthetic post-acetazolamide CBF maps from baseline ASL MRI, avoiding drug administration in high-risk stroke patients. A U-Net–based conditional autoencoder outperformed three diffusion models on 194 paired scans and accurately captured clinically relevant CVR patterns, demonstrating a practical, data-efficient approach for drug-free CVR assessment.

**Strengths:**

The paper is motivated by a clinical need to replace invasive acetazolamide-based CVR assessment with a safe, non-invasive alternative. It provides a comparison showing that a simple U-Net–based conditional autoencoder outperforms more complex diffusion models, challenging the assumption that higher model complexity always yields better medical imaging performance. The evaluation goes beyond pixel-level metrics by demonstrating clinically meaningful CVR patterns, increasing translational relevance. Limitations and future directions are transparently discussed, and the study is well-structured, clearly written, and reproducible. Overall, it offers scientific merit and practical value to the medical imaging community.

**Weaknesses:**

The study is limited by its reliance on 2D mid-axial slices, which undermines whole-brain CVR assessment and limits clinical applicability. Its single-center dataset of modest size lacks external validation, raising concerns about generalizability and site-specific bias. Clinical validation is insufficient, as the work stops at image similarity and regional analysis without demonstrating diagnostic accuracy or outcome relevance. The poor performance of diffusion models is not deeply analyzed, leaving uncertainty about whether this reflects model limitations or suboptimal tuning. Finally, the percent-change analysis is methodologically weak, relying on unstable voxel-wise calculations and arbitrary capping, which may distort physiological interpretation.

**Detailed Comments:**

The caption for Figure 1 is sparse. It should explicitly state what the error bars represent and indicate the sample size (n) for each disease laterality group within the figure or caption. The y-axis label "% CBF Change" should specify if this is the mean percent change.
 In Table 1, consider adding a column for the standard error of the mean (SEM) alongside the standard deviation, as the SD is large relative to the mean for some metrics. This would help readers better gauge the precision of the performance estimates.
The description of the K-fold cross-validation for hyperparameter tuning is vague. Specify the value of K and briefly mention which hyperparameters were searched. This enhances reproducibility.
The discussion on why the cAE outperformed diffusion models is insightful but could be deepened. Briefly hypothesize why the inductive bias of the U-Net is better suited. Is it because the L1+SSIM loss directly optimizes for the task's needs, whereas the diffusion objective is a more indirect proxy for this specific, constrained mapping?
When first mentioning the "steal"-like decreases (page 8), add a brief explanatory phrase or citation for a broader audience.
The limitations are appropriately listed in the conclusion. For greater impact, consider introducing the most critical limitations earlier in the discussion section when interpreting the results, framing them as boundaries on the generalizability of the current findings.
The suggestion to adopt more robust regional statistics is excellent. Provide one or two concrete examples in the text.
The sentence fragment "...tailored to the" at the bottom of page 3/start of page 4 appears to be a formatting error and should be connected to the beginning of Section 3 for coherence.
For Figure 2, it would be highly informative to add a fourth panel showing the absolute error map or the percent change map for both ground truth and prediction. This would visually substantiate the claimed high fidelity and capture of subtle changes.
 In the abstract and conclusion, the paper's key contribution could be more sharply framed: not just "a model that predicts post-ACZ CBF," but "an empirical demonstration that a deterministic conditional autoencoder outperforms more complex generative models for this specific clinical prediction task, offering a simpler, high-fidelity alternative for drug-free CVR assessment."  This more precisely captures the paper's valuable insight.

**Justification Of The Preliminary Rating:**

The paper sits on the borderline between acceptance and rejection due to a clear trade-off between strong methodological insight and limited clinical validation. Its key contribution is showing that a simple deterministic cAE outperforms more complex diffusion models for hemodynamic mapping is valuable and challenges the assumption that greater model complexity yields better performance. The experimental comparison is fair, well-executed, and addresses an important clinical problem.
However, the impact is constrained by major limitations. The use of a single 2D slice oversimplifies a whole-brain physiological task, weakening clinical relevance. Validation is limited to image fidelity, with no demonstration of diagnostic or decision-level equivalence to real ACZ challenges, and no external or robust internal validation to support generalizability. As a result, the work makes a strong ML contribution but a weak clinical one. Acceptance depends on whether the venue prioritizes methodological insight or clinically validated medical imaging, addressing clinical validation and generalizability in rebuttal would be essential to tip the balance toward acceptance.

**Questions To Address In The Rebuttal:**

Authors must convincingly justify the use of a single 2D mid-axial slice, supported by evidence showing how much of the brain or key vascular territories it represents and whether slice-based CVR measures meaningfully correlate with whole-brain 3D CVR or clinical outcomes in Moyamoya disease. Without this, the clinical relevance remains weak.
The authors must address generalizability.
Results from an external validation cohort are strongly needed; if unavailable, they should provide a concrete plan and commitment to obtain such data. At a minimum, more rigorous internal validation should be presented to demonstrate robustness.
The paper must establish clinical, not just image-level, validity. The rebuttal should include a clear diagnostic evaluation framework, such as showing that synthetic post-ACZ maps can classify impaired versus preserved CVR with meaningful accuracy.
The authors must also explain why the reduced dynamic range in predicted CVR changes would not compromise clinical decision thresholds.
The failure of diffusion models requires deeper analysis. The authors should demonstrate thorough experimentation through ablation or sensitivity studies and discuss whether limited dataset size inherently disadvantages diffusion models compared to a deterministic cAE.
Overall, it must go beyond promises of future work and provide new analyses, evidence, or concrete validation plans. Without this, the study remains a preliminary proof-of-concept with limited clinical relevance and insufficient methodological validation.

---

> ### Author Response · Authors · 2026-01-25
>
> Thank you for the detailed critique and concrete suggestions. We address: (i) justification of a single 2D slice, (ii) clinical applicability of the evaluation, (iii) generalizability and external validation, (iv) diffusion-vs-deterministic interpretation, and (v) the percent-change methodology.
>
> (1) Justification for using a single 2D mid-axial slice and clinical relevance: We agree that whole-brain 3D CVR assessment would be ideal. In Moyamoya, however, CVR-relevant hemodynamics primarily involve the anterior circulation, especially the bilateral MCA territories, which are consistently intersected by the mid-axial plane. This slice captures regions most relevant to CVR interpretation, lateralized impairment, and vascular steal, while posterior/inferior regions are often less informative. Moreover, CVR interpretation is commonly performed at the vascular-territory level, so full 3D synthesis is not strictly required to test whether the model preserves key clinical signatures. Methodologically, restricting to a representative 2D slice improves stability and data efficiency in a limited paired-data regime (194 paired scans), enabling a clean feasibility test of whether baseline ASL contains learnable signal for post-ACZ response. We emphasize this is a 2D, single-center feasibility study and not a quantitative substitute for full 3D CVR assessment.
>
> (2) Why the evaluation probes clinical applicability (not only image similarity): We agree that image-level metrics alone are insufficient to prove clinical relevance. The evaluation is intentionally paired: (i) MAE/SSIM/PSNR quantify fidelity and reduce the risk of hallucinated structure, and (ii) ROI analyses compute percent CBF change within atlas-defined MCA territories, stratified by disease laterality, testing whether the model preserves clinically meaningful CVR signatures (blunted augmentation in affected territories and steal-like paradoxical decreases relative to contralateral/healthier regions). Although the model compresses ΔCBF magnitude, it consistently preserves the territorial ordering; this reflects a conservative bias that lowers the risk of falsely reassuring predictions in a high-stakes setting and supports the intended role as decision-support/screening rather than definitive replacement of pharmacologic CVR testing. Accordingly, we do not advocate applying absolute ΔCBF thresholds directly to synthetic maps without calibration; any thresholding should be calibrated on a validation set prior to downstream use.
>
> (3) Generalizability and external validation: We agree that external validation is the gold standard. Paired pre/post-ACZ ASL cohorts in Moyamoya are rare and difficult to assemble, and the present evidence is limited to a single-center cohort collected under strict privacy and infrastructure constraints; claims should be interpreted as single-center feasibility with a held-out test set rather than multi-site generalization. Code/implementation release is the most practical mechanism to enable independent evaluation at centers with comparable cohorts, and multi-center validation remains the appropriate next step for clinical translation. We are in communication with other research centers to establish such collaborations while adhering to data protection regulations.
>
> (4) Diffusion underperformance (limitation vs tuning, and how narrowly to interpret the conclusion): Our conclusions about diffusion are intentionally bounded: among the evaluated conditional diffusion formulations, diffusion underperformed the deterministic cAE on fidelity metrics and preserved territorial physiological patterns less consistently in this limited paired-data, low-SNR mapping regime. This is not intended as an exhaustive statement about all diffusion variants. A plausible explanation is objective alignment and inductive bias: the cAE is directly optimized for fidelity to the target (L1+SSIM), whereas diffusion training optimizes an indirect denoising objective that may be less well matched to this constrained physiological mapping when data are limited.
>
> (5) Percent-change analysis and low-baseline instability: We agree voxel-wise percent change is sensitive to near-zero baseline CBF. However, in Moyamoya, very low baseline perfusion can reflect true severe hypoperfusion; excluding low-baseline voxels as “outliers” would systematically remove the most pathologic tissue and bias territory summaries toward healthier regions, potentially masking clinically relevant impairment and steal-like behavior. The analysis therefore reflects an explicit bias–variance tradeoff inherent in voxel-wise normalization in low-SNR ASL and is best interpreted primarily through territory-level patterns rather than individual extreme voxels.
>
> (6) Reproducibility/clarity updates: We added methodological detail in the manuscript (hyperparameter selection and model implementation) and clarified reporting in Fig. 1/Table 1 and key terminology for readability and reproducibility.

---

> > ### Comment · Reviewer_s8hq · 2026-02-02
> >
> > I sincerely appreciate the authors for their considerable efforts and for the clear, structured, and thoughtful manner in which they have addressed the assigned comments. The responses are well-articulated and demonstrate a genuine commitment to improving the clarity, quality, and rigor of the manuscript.

---

### Official Review · Reviewer_stsC · 2026-01-09

**Confidence:** 3
**Preliminary Rating:** 3

**Summary:**

CVR is typically assessed by administering acetazolamide (ACZ) and acquiring post-ACZ perfusion maps. This is expensive. The authors use deep learning models to predict post-ACZ perfusion directly from baseline arterial spin labeling (ASL) MR. The show on a held out test set, the proposed model outperforms baselines in terms of reconstruction fidelity.

**Strengths:**

the authors addressed a key clinical need by giving non-invasive CVR assessment using MRI. The proposed model is simple and effective. The evaluation involves multiple generative methods, including DDPM, residual diffusion, and cold diffusion.

**Weaknesses:**

the proposed method only involves a 2d middle axial slice, rather than the full 3D volume. We are throwing away a lot of information when extracting this 2d slice from the 3d volume. Results are only on one single dataset with 194 patients, limiting the generalizability of the findings.

**Detailed Comments:**

" For left-sided
disease (n = 11), ground truth showed larger CBF increases in healthy versus diseased MCA
territories (67.41% unhealthy vs. 223.29% healthy), and the cAE preserved this pattern
with a compressed dynamic range (30.60% vs. 69.83%). The same ordering was observed
for right-sided disease (47.03% vs. 131.22% in ground truth, 7.10% vs. 97.19% for the cAE)
and bilateral disease (136.38% vs. 1308.48% in ground truth, 42.78% vs. 206.72% for the
cAE), indicating that the model consistently learned that healthy MCA territories should
exhibit larger perfusion augmentation than their diseased counterparts."

The model underestimates CBF increases. how does it affect high stake clinical decision making?

**Justification Of The Preliminary Rating:**

The authors provide a robust comparison between deterministic and generative architectures, offering a valuable finding that an autoencoder can outperform diffusion models in this specific low-data medical imaging regime. Furthermore, the model's ability to recover clinically relevant physiological patterns demonstrates its potential for diagnostic utility.

**Questions To Address In The Rebuttal:**

please see weakness above. could the proposed method be extended to the full 3d volume instead of just a 2d slice?

---

> ### Author Response · Authors · 2026-01-25
>
> Thank you for the constructive feedback and for raising two key issues: (i) using only a 2D slice rather than the full 3D volume, and (ii) the clinical implications of dynamic-range underestimation.
>
> (1) Why a single middle axial slice is clinically meaningful, and why 3D is not strictly necessary for the specific clinical question tested here: We agree that full 3D prediction is a natural long-term goal. The current submission is explicitly positioned as a proof-of-concept feasibility study under limited paired pre/post-ACZ data (194 paired scans), and the mid-axial slice is chosen as a clinically motivated plane rather than a generic proxy. In Moyamoya, cerebrovascular pathology predominantly involves the anterior circulation, particularly the bilateral MCA territories, which are consistently intersected by the middle axial plane. This slice captures the regions most relevant to CVR-based clinical interpretation, assessment of lateralized impairment and vascular steal, while excluding posterior and inferior regions that are typically less informative for Moyamoya-specific CVR assessment. Consistent with clinical practice, interpretation is driven primarily by vascular-territory–level patterns rather than requiring voxel-complete 3D coverage for the intended signatures evaluated in this work. From a methodological standpoint, restricting the problem to a representative 2D slice improves data efficiency and model stability in a limited-data regime, enabling a rigorous test of the core question: whether post-ACZ hemodynamic changes are predictable from baseline ASL at all.
>
> (2) Underestimation of CBF increases and high-stakes clinical decision making: We agree that dynamic-range compression is a limitation, especially if one were to apply absolute thresholds without additional calibration and validation. The evaluation is therefore designed to probe clinical applicability at the level most relevant to CVR interpretation in steno-occlusive disease: whether affected MCA territories show blunted or paradoxical responses relative to contralateral/healthier regions. In that respect, despite magnitude compression, the model consistently preserves clinically expected territorial structure: larger augmentation in healthier tissue than diseased tissue and steal-like decreases in high-risk regions. This behavior is conservative in a clinically important way, preserving ordering while compressing magnitude reduces the risk of producing falsely reassuring exaggerated improvements, and supports positioning the method as a screening/decision-support proof-of-concept rather than a replacement for pharmacologic CVR testing in definitive decision-making.

---

### Official Review · Reviewer_ULsB · 2026-01-11

**Confidence:** 5
**Preliminary Rating:** 5
**Final Rating:** 5

**Summary:**

This paper explores whether deep learning can generate post-acetazolamide (post-ACZ) cerebral blood flow MRI directly from baseline ASL scans, eliminating the need for pharmacological challenges in high-risk stroke patients. The authors propose a conditional autoencoder (cAE) as the primary model and compare it against diffusion-based alternatives. cAE achieves the best reconstruction fidelity (highest SSIM, PSNR, lowest MAE), outperforming diffusion models that introduced noise and blur. The significance lies in demonstrating the feasibility of non-invasive CVR assessment, potentially reducing reliance on ACZ administration and expanding access to safer stroke risk evaluation.

**Strengths:**

The paper addresses a pressing need in cerebrovascular imaging—reducing reliance on acetazolamide challenges for CVR assessment. By proposing a computational alternative, it offers a safer and more accessible workflow for high-risk stroke patients, which is highly valuable to the medical community.
The study uses a reasonably sized dataset (194 paired scans) with a clear train/validation/test split, and evaluates performance using multiple complementary metrics (SSIM, PSNR, MAE) alongside clinically meaningful MCA territory analyses. This dual validation adds credibility to the findings.

**Weaknesses:**

The choice to reduce 3D ASL volumes to a single middle axial slice improves computational efficiency but sacrifices spatial context. This simplification may limit the model’s ability to capture perfusion heterogeneity across the full brain, which is clinically relevant in Moyamoya and other cerebrovascular disorders.

**Detailed Comments:**

The diffusion-based baselines are relatively simple. It would be useful to acknowledge that more advanced conditional diffusion frameworks (e.g., latent diffusion or transformer-based denoisers) could yield stronger results, and position the cAE as a practical baseline rather than dismissing diffusion outright.

**Justification Of Final Rating:**

it makes a clinically meaningful and technically sound contribution. Rreducing reliance on acetazolamide challenges for cerebrovascular reserve assessment, which is costly, time-consuming, and contraindicated for some patients. Their proposed conditional autoencoder (cAE) demonstrates that a relatively simple, data-efficient architecture can outperform more complex diffusion-based generative models, achieving superior reconstruction fidelity and preserving clinically relevant perfusion patterns.

**Justification Of The Preliminary Rating:**

it makes a clinically meaningful and technically sound contribution. Rreducing reliance on acetazolamide challenges for cerebrovascular reserve assessment, which is costly, time-consuming, and contraindicated for some patients. Their proposed conditional autoencoder (cAE) demonstrates that a relatively simple, data-efficient architecture can outperform more complex diffusion-based generative models, achieving superior reconstruction fidelity and preserving clinically relevant perfusion patterns.

**Questions To Address In The Rebuttal:**

.

---

> ### Author Response · Authors · 2026-01-25
>
> Thank you for the supportive and thoughtful review. We appreciate your emphasis on the clinical motivation: reducing reliance on acetazolamide (ACZ) challenge imaging for CVR assessment in high-risk patients where ACZ can be costly, time-consuming, contraindicated, or poorly tolerated.
>
> (1) 2D slice vs. 3D volumetric modeling: We agree that reducing 3D ASL volumes to a single slice sacrifices spatial context. The middle axial slice is used here as a clinically motivated plane rather than merely a computational shortcut: Moyamoya pathology and CVR-relevant hemodynamics predominantly involve the anterior circulation, particularly the bilateral MCA territories, which are consistently intersected by the middle axial plane. This slice therefore captures the regions most relevant for CVR-based interpretation, including lateralized impairment and steal-like (negative CVR) behavior, while posterior/inferior regions are often less informative for this disease context. Importantly, CVR interpretation in clinical practice is typically performed at the vascular-territory level, so voxel-complete 3D coverage is not strictly required to test the specific clinical signatures evaluated in this proof-of-concept.
>
> (2) Diffusion baselines and scope of the conclusion: We agree that additional diffusion-based methods could be explored in future work. However, we would emphasize that we compare four different generative approaches in our work. Our conclusions are intentionally bounded to the specific conditional diffusion formulations tested here. The empirical result is that, in this limited paired-data, low-SNR physiological mapping regime, the deterministic cAE yields superior fidelity (MAE/SSIM/PSNR) and more consistently preserves clinically relevant territorial perfusion patterns than the evaluated diffusion alternatives; this should not be interpreted as a general statement that all conditional diffusion frameworks (e.g., latent diffusion or transformer denoisers) are unsuitable for the task.

---

### Author Rebuttal · Authors · 2026-01-25

**Rebuttal:**

We thank the reviewers for their constructive feedback and for underscoring the clinical motivation. Regarding the 2D slice setting (stsC, s8hq, ULsB), we agree 3D is a long-term goal; the mid-axial slice was a deliberate, clinically motivated choice that also improves data efficiency. In Moyamoya, CVR-relevant pathology predominantly involves the anterior circulation (bilateral MCA territories), which is consistently intersected by the mid-axial plane. This slice captures the primary CVR biomarkers used in practice (lateralization and steal-like responses), while posterior/inferior regions are typically less informative. Restricting to 2D improves stability in a limited paired-data regime ($N=194$), enabling a focused feasibility test of whether baseline ASL contains learnable signal for post-ACZ response. Regarding clinical utility and dynamic range (stsC), while the cAE compresses the absolute magnitude of $\Delta$CBF, it preserves the critical territorial ordering (healthier tissue shows larger augmentation than diseased tissue) and preserves steal-like (paradoxical) decreases in high-risk regions. This conservative behavior supports screening/decision-support by reducing the risk of falsely reassuring exaggerated increases while maintaining sensitivity to pathology. For the percent-change critique (s8hq), voxel-wise ratios are sensitive to very low baseline CBF; however, thresholding low-baseline voxels as “outliers” would preferentially exclude severely hypoperfused (pathologic) tissue. We therefore retain low-baseline voxels and transparently document the resulting bias–variance tradeoff when summarizing changes within MCA territories. For diffusion vs. deterministic models (ULsB, s8hq), conclusions are intentionally bounded to the evaluated conditional diffusion formulations; a likely contributor is objective alignment/inductive bias, with the cAE’s direct L1+SSIM fidelity objective better matched to low-SNR physiological regression than denoising-based training. Finally, the revised manuscript improves reproducibility and clarity: Fig. 1 specifies subgroup sizes and SD error bars; Table 1 reports mean$\pm$SD on the 49-patient test set; Methods details K-fold selection and validation-based model selection and defines “steal-like” behavior at first mention. We will release code to facilitate external validation and broader use.

**Supporting Material:**

/attachment/bed21a18f0292fb9d2f576bb82cbeb68e574ec7f.pdf

---

### Comment · Area_Chair_fhfV · 2026-02-02

Dear Reviewers,
We kindly ask you to carefully read the authors’ rebuttals, if not done so already, before finalizing your review.
As we approach the final stage of the review process, please update your Final Rating for each assigned paper by navigating to:
“Edit” → “Official Review” → Final Rating. Kindly complete this update by February 1st, 2026 (23:59 AoE).
Thank you very much for your continued effort and valuable contributions.

---

### Meta-Review · Area_Chair_fhfV · 2026-02-09

**Recommendation:** Accept (Poster)
**Confidence:** 4

**Metareview:**

This paper presents a feasibility study for predicting post ACZ CVR maps directly from baseline ASL using a conditional autoencoder. It is compared against a couple of baseline diffusion-based methods. All reviewers agreed that the clinical motivation is strong and that the study is sound.

The reliance on a single 2D mid axial slice rather than full 3D data was raised by all reviewers but is appropriate given the framing of the work as a proof-of-concept, and is still clinically relevant. The performance of the cAE as compared to the diffusion models is also well justified in the rebuttal.

The underestimation of CBF change is still a concern and should be further evaluated along with the multi-center generalizability, in future work.

Nevertheless, this is a solid, well-supported feasibility study that yields valuable insight so I recommend acceptance.

---

### Decision · Program_Chairs · 2026-02-13

Accept (Poster)